# EspP2 Regulates the Adhesion of *Glaesserella parasuis* via Rap1 Signaling Pathway

**DOI:** 10.3390/ijms25084570

**Published:** 2024-04-22

**Authors:** Xinwei Tang, Shiyu Xu, Zhen Yang, Kang Wang, Ke Dai, Yiwen Zhang, Bangdi Hu, Yu Wang, Sanjie Cao, Xiaobo Huang, Qigui Yan, Rui Wu, Qin Zhao, Senyan Du, Xintian Wen, Yiping Wen

**Affiliations:** Research Center of Swine Disease, College of Veterinary Medicine, Sichuan Agricultural University, Chengdu 611130, China

**Keywords:** *Glaesserella parasuis*, EspP2 protein, adhesion, Rap1 signaling pathway

## Abstract

Different levels of EspP2 expression are seen in strains of *Glaesserella parasuis* with high and low pathogenicity. As a potential virulence factor for *G. parasuis*, the pathogenic mechanism of EspP2 in infection of host cells is not clear. To begin to elucidate the effect of *EspP2* on virulence, we used *G. parasuis* SC1401 in its wild-type form and SC1401, which was made *EspP2*-deficient. We demonstrated that EspP2 causes up-regulation of claudin-1 and occludin expression, thereby promoting the adhesion of *G. parasuis* to host cells; *EspP2*-deficiency resulted in significantly reduced adhesion of *G. parasuis* to cells. Transcriptome sequencing analysis of EspP2-treated PK15 cells revealed that the Rap1 signaling pathway is stimulated by EspP2. Blocking this pathway diminished occludin expression and adhesion. These results indicated that EspP2 regulates the adhesion of *Glaesserella parasuis* via Rap1 signaling pathway.

## 1. Introduction

Gram-negative *Glaesserella parasuis* (*G. parasuis*) is a member of the *Pasteurella* family of bacteria that is dependent on nicotinamide adenine dinucleotide (NAD) [1]. It is an opportunistic parasitic pathogen that can cause a condition known as Glässer’s disease; symptoms include polyserositis, arthritis, and sepsis [2]. There are now 15 serovars of *G. parasuis* known, ranging in virulence from highly virulent to nonvirulent [3]. To prevent *G. parasuis* infection of pigs, vaccination and antibiotic therapy are the most widely utilized methods. However, because of their low protective efficacy or poor cross-protection vaccine, failures are frequently reported [4]. Additionally, clinical isolates of *G. parasuis* exhibit resistance to several antibiotics [3]. Accordingly, exploring additional therapeutic and prophylactic methodologies is required.

Pathogen adhesion to host cells is crucial to the infection and pathogenesis processes. *G. parasuis* binds to a variety of host cells, including porcine brain capillary endothelial cells [5], porcine kidney epithelial cells (PK15 cells) [6], and porcine umbilical vein endothelial cells [7], among others. However, the precise mechanisms of adhesion are still unknown.

With more than 700 members, the autotransporter protein (AT) family is the largest family of extracellular proteins found in Gram-negative bacteria [8,9]. These proteins are synthesized as precursor proteins with three common functional domains: a passenger domain, an N-terminal signal peptide, and a C-terminal translocator domain that forms a pore in the outer membrane to help transport the internal passenger domain to the bacterial surface [10,11]. Glycosylated adhesin AIDA-1 of *E. coli* is a classical Va-type autotransporter; it adheres to the epithelial cells of various animals by interacting with receptors, and it plays a significant role in virulence [12]. YadA of *Yersinia pseudotuberculosis* is a trimeric autotransporter. In the intestine, YadA-mediated adhesion is essential for invasion and spread. YadA not only acts as an adhesin, it can also promote bacterial entry into host cells [13]. In this investigation, we looked into the function of a putative extracellular serine protease, EspP2, and investigated the role of this AT protein in *G. parasuis* adhesion. EspPα is a subtype of EspP in enterohemorrhagic *Escherichia coli* (EHEC) strains. The discovery that the pO157 plasmid, which contains EspPα, is required for *E. coli* O157:H7 to completely adhere to HEp-2 epithelial cells [14] and affects the colonization of the bovine terminal rectum [15], provided the initial impetus to examine the possibility of EspPα contributing to cellular adherence.

Hosts need an intact epithelial barrier to protect against pathogen infection. Tight junctions (TJ) close the intercellular space, which is key to the formation of the epithelial barrier [16]. Of the proteins specifically localized to tight junctions, claudin and occludin proteins are major types [17]. Numerous pathogens that affect epithelial cells have developed complex tactics to get past this defense and spread their infections. It is, however, unknown which precise mechanism *G. parasuis* uses to adhere to epithelial cells. Growing evidence has demonstrated that different pathogens influence different tight junction components to induce infection. Luo et al. [18] reports that the knockout of occludin leads to a decrease in Porcine Epidemic Diarrhea Virus infection, and the tight junction proteins claudin-1 [19] and occludin [20] are necessary for the entry of the Hepatitis C virus.

A small GTPase from the Ras family, Rap1 is engaged in several signal transduction pathways within cells. For instance, Rap1 is required for intracellular bacterial pathogens to form their replication-permissive vacuole [21]. The Rap1 signaling pathway is significantly activated in *G. parasuis*-infected porcine aortic vascular endothelial cells [22]. The activation of NF-κB in intestinal epithelial cells induced by heat-labile enterotoxins is dependent on the activation of the Ras-like GTPase Rap1 in a cAMP-dependent manner, which in turn promotes the adherence of enterotoxigenic *E. coli* [23]. Caseinolytic protease L regulates the adhesion of *Streptococcus pneumococcal* to A549 human lung cells by inducing and activating Rap1 [24].

Here, we investigated the function of EspP2 in *G. parasuis* adherence to epithelial cells. To this end we constructed an *EspP2* deficient strain of *G. parasuis* and compared its adherence on PK15 cells with wild-type *G. parasuis*. Additionally, PK15 cells were treated with pure EspP2, and RT-PCR, WB, and indirect immunofluorescence were used to assess the expression levels of occludin and claudin-1. To further evaluate the effect of *EspP2* on *G. parasuis* adhesion, we examined the adhesion of Δ*EspP2*::Kan and wild-type *G. parasuis* on four cells lines (claudin-1 knockout, occludin knockdown, claudin-1 overexpressing, and occludin overexpressing). By transcriptomic analysis of PK15 cells incubated with purified EspP2, we found that EspP2 activates the Rap1 signaling pathway. This study advances our knowledge of the mechanisms of *G. parasuis* adhesion and provides a theoretical framework for the creation of novel medications to treat *G. parasuis* infection.

## 2. Results

### 2.1. Deletion of EspP2 Decreases G. parasuis Adherence

An *EspP2* deletion strain of *G. parasuis* was constructed by transforming SC1401 with the suicide plasmid pK18-*EspP2* containing fragments homologous to *EspP2* (Figure 1A). DNA from the putative *EspP2* deletion and wild-type SC1401 strain was amplified using HPS, *EspP2*, and Kan primers. Electrophoresis of the PCR products demonstrated the proper construction of Δ*EspP2*::Kan (Figure 1B).

As shown in Figure 2A, no discernible variations in growth were observed between Δ*EspP2*::Kan and wild-type SC1401 in vitro. This result eliminates the possibility that the attenuation or enhancement of toxicity may be due to suppressed or increased metabolic levels in Δ*EspP2*::Kan. To begin investigating the effect of Δ*EspP2* on *G. parasuis* interactions with PK-15 cells, a comparison was made between the adhesion ability of wild-type and mutant strains. As illustrated in Figure 2B, the adherence of Δ*EspP2*::Kan (tested at MOI 100 and 10) was significantly lower than that of wild-type SC1401. Thus, the loss of *EspP2* results in reduced bacterial adhesion PK-15 epithelial cells. We also ran the same experiment with new-born piglet tracheal (NPTr) cells. The results showed that deletion of *EspP2* resulted in reduced bacterial adhesion to NPTr epithelial cells, consistent with that in PK-15 (Appendix A).

### 2.2. EspP2 Enhances Expression of Claudin-1 and Occludin

To investigate the role of *EspP2* on adhesion of *G. parasuis* to PK15 cells, EspP2 with 6×His tags on its N-terminal was expressed *G. parasuis* (Appendix A). To exclude the influence of the His-tag on the results, the His-tag protein was expressed and purified as a control protein (Appendix A). Claudin-1 and occludin expression levels were not significantly impacted by the His-tag protein, as confirmed by qRT-PCR and WB (Appendix A). The mRNA (Figure 3A) and protein levels (Figure 3B) of occludin and claudin-1 were significantly elevated in PK15 cells treated with pure EspP2-His at every time point measured from 6 to 48 h. Immunostaining showed similar results (Figure 3C,D). To further understand the relationship of EspP2 to the expression of claudin-1 and occludin, PK-15 cells were infected with wild-type SC1401 and Δ*EspP2*::Kan. In cells infected with wild-type SC1401, the levels of claudin-1 and occludin increased through 12 h and 36 h of infection, respectively, before decreasing to time 0 levels or lower (Figure 3E). In cells infected with Δ*EspP2*::Kan, levels of occludin and claudin-1 decreased significantly throughout the 48 h infection (Figure 3F). These findings showed that occludin and claudin-1 expression are impacted by EspP2.

### 2.3. EspP2 Regulates Adhesion by Affecting Claudin1 and Occludin

Five cell lines (PK15, claudin-1 knockout, occludin knockdown, claudin-1 overexpressing, and occludin overexpressing) were incubated with either wild-type SC1401 or Δ*EspP2*::Kan at MOI 10 and 100 for 2 h. The extent of bacterial adhesion, tested in PK15, claudin-1 knockout, and occludin knockdown cells, was accessed by colony counting. Figure 4A,B show that, on PK15 cells, there were significantly fewer Δ*EspP2*::Kan CFUs produced than wild-type CFUs; (6.5 × 10^4^ CFU/well vs. 3.73 × 10^5^ CFU/well respectively at MOI 100). On claudin-1 knockout and occludin knockdown cells, there were also significantly fewer CFUs of both bacteria, compared to PK15 cells. Figure 4C shows Giemsa staining of PK15, claudin-1 knockout, and occludin knockdown cells incubated with wild-type and Δ*EspP2*::Kan *G. parasuis*.

The adhesion of both wild-type and Δ*EspP2*::Kan was, as expected, significantly greater on claudin-1 and occludin overexpressing cells than on PK15 cells (Figure 5A–C). These results show that EspP2 affects levels of claudin-1 and occludin, which in turn influences *G. parasuis*’s capacity to adhere to PK-15 cells.

### 2.4. EspP2 Treatment Activates the Rap1 Signaling Pathway

To investigate the interaction mechanisms between claudin-1, occludin, and EspP2, we conducted a comparative transcriptomic study on PK15 cells incubated with purified EspP2 for 0, 12, and 36 h. There were three technical duplicates run. Each sample had Total Mapped Reads of greater than 70% and Multiple Mapped Reads of less than 5%, demonstrating that the reference genome chosen for this experiment was suitable and that the associated experiments were uncontaminated (Appendix A). For every biological replicate, the squared Pearson correlation coefficient (R^2^) was higher than 0.92, indicating that the replication between samples was good (Appendix A). Six genes with up-regulated transcription levels and six with down-regulated transcription levels were arbitrarily chosen for relative fluorescence quantification in order to validate the transcriptome sequencing results. The results of the fluorescence quantitation are consistent with the trend in gene expression in transcriptome sequencing results (Appendix A), demonstrating the validity of the RNA-Seq data and their suitability for further investigation.

Transcriptome analysis showed that, compared with the 0 h control, there were 1734 differentially expressed genes after 12 h incubation. Of these, 1002 were down-regulated and 732 were up-regulated (Figure 6A). After 36 h incubation, there were 1367 differentially expressed genes. Of these, 684 were down-regulated and 683 were up-regulated (Figure 6B), indicating that *EspP2* affects multiple biological functions in *G. parasuis*. KEGG pathway analysis and GO categorization were applied to the DEGs. According to GO findings, the apical part of cell, apical plasma membrane, cell communication, and signal transduction were the most prevalent categories (Figure 6C). KEGG analysis revealed that the significantly enriched pathways were cytokines and cytokine receptors, TNF signaling pathway, MAPK signaling pathway, PI3K-Akt signaling pathway, IL-17 signaling pathway, and Rap1 signaling pathway (Figure 7A). Of these, the Rap1 signaling pathway was one of the 20 most represented pathways at both 12 and 36 h incubation. The Rap1 signaling pathway is activated in PK15 cells treated with EspP2, as seen by the markedly elevated transcriptional expression of *RAP1B* in Figure 7B.

### 2.5. Inhibition of the Rap1 Signaling Pathway Reduces Occludin Expression and Inhibits Adhesion

To ascertain if the expression of occludin and claudin-1 is impacted by EspP2’s stimulation of the Rap1 signaling pathway, we treated PK15 cells with 1 μM ESI-05, an inhibitor of the Rap1 signaling pathway, and then incubated them with purified EspP2. Occludin levels, as assessed by Western blotting, were significantly lower in Rap1-inhibited cells treated with EspP2 than in uninhibited cells treated with the same substance (Figure 8A). Adherence of wild-type and Δ*EspP2*::Kan *G. parasuis* to treated PK15 cells was significantly decreased over their adherence to untreated PK15 cells, demonstrating that inhibition of Rap1 signaling results in reduced *G. parasuis* adhesion (Figure 8B). Figure 8C shows Giemsa staining of treated and untreated PK15 cells incubated with wild-type and Δ*EspP2*::Kan *G. parasuis*.

## 3. Discussion

The genome of *G. parasuis* SC1401 contains two EspP proteins, EspP1 and EspP2; their amino acid sequences are 72.4% homologous and their gene sequences are 76.9% homologous. Wang et al. have reported that EspP2 in *G. parasuis* exhibits more protease activity than EspP1 [26], indicating that EspP2 may play a stronger role in virulence. Here, we studied *G. parasuis* EspP2 using a complete recombinant protein expressed in *E. coli*.

Autotransporter proteins can secrete their domains through the outer membrane of Gram-negative bacteria, and are associated with infection and virulence. Similar to the known cleavage sequence of EspP in *E. coli*, EVNNLN, the EspP protein of *G. parasuis* contains a sequence called EMNNLN [11]. After lysis, the passenger domain of EspP from *E. coli* is released into the extracellular environment, which is why it is found in culture supernatants rather than precipitates post-lysis. In contrast, EspP from *G. parasuis* is found in the bacterial precipitate post-lysis [27]. Therefore, more research is needed to determine whether the suggested cleavage sequence of *G. parasuis* EspP is an active site.

To date, gene knockout strains of *G. parasuis* can be made by natural transformation, conjugal transfer, and electro-transformation. Conjugal transfer depends on the presence of sexual fimbriae and conjugal plasmid of *E.coli*, and electro-transformation is an artificial transformation method; compared with these methods, natural transformation does not depend on other mobile elements [10]. Although natural transformation is dependent on whether the host bacteria are naturally competent and the frequency of natural transformation, it is a convenient and efficient method for transforming *G. parasuis* SC1401.

We concluded that *EspP2* deletion reduces the adhesion capacity of *G. parasuis* to epithelial cells in both PK-15 and NPTr cell lines. However, in subsequent experiments, we used only PK-15 cells. The initial adhesion results of NPTr and PK-15 cells were comparable and both are epithelial cell lines, but PK-15 cells are the more commonly used as they are more mature and stable.

In this study we observed that occludin and claudin-1 expression levels are up-regulated by EspP2. In many cases, the expression of tight junction proteins is reduced in cells infected with pathogens [28,29,30,31]. The mechanism by which EspP2 promotes claudin-1 and occludin up-regulation is unknown. Two possibilities are proposed here. First, tight junction structure maintenance and increased production of tight junction proteins are necessary to preserve the integrity of the epithelial barrier [32,33]. Second, tight junction protein expression is significantly influenced by the inflammatory mediators that cells release following EspP2 activity. [34,35,36].

In order to explore the effect of claudin-1 and occludin on bacterial adhesion, *claudin-1* knockout or *occludin* knockdown PK15 cells were used to compare the differences in adhesion between wild-type and Δ*EspP2*::Kan *G. parasuis* SC1401. Bacterial adhesion was greatly enhanced in overexpressed cells and decreased in deletion cells. Studies have indicated that integrins play a crucial role in numerous host–pathogen interactions, including pathogenic bacteria, viruses, and fungi [37]. Adhesins found in most bacteria have the ability to attach to integrins either directly or indirectly [37], and TJs are structurally related to integrins [38]. The up-regulation of claudin-1 and occludin by EspP2 contributes to the adhesion of *G. parasuis*. It is possible that EspP2 itself is an adhesin that interacts with tight junction proteins to mediate bacterial adhesion. It may also directly affect the expression of integrins on the surface of host cells. Whether TJs directly or indirectly mediate the adhesion of *G. parasuis* needs to be further studied.

The small GTPase Rap1 is either in a GDP-bound inactive state, or a GTP-bound active state. It can function as a molecular switch to regulate a variety of cellular processes, such as the formation of intercellular junctions, thanks to the cycle between the two states [34]. Previous studies have demonstrated that Rap1 regulates the formation of E-cadherin-mediated adhesion junctions in Drosophila and mammalian epithelial cells [39,40]. Rap1 activation also plays a role in endothelial cells’ ability to establish tight junctions [41]. Thus, we investigated whether tight junction protein expression is impacted by EspP2 activation of the Rap1 signaling pathway. We found that inhibiting Rap1 by ESI-05, an inhibitor of EAPC2 which mediates Rap1 activation, reduced occludin expression in PK15 cells. This demonstrated a favorable correlation between the activation of the Rap1 pathway and occludin expression. As an important component of tight junctions, occludin can bind to actin cytoskeleton [42]. Among the signaling proteins involved in junctional regulation, small GTPases are of particular interest due to their ability to cycle between active and inactive states. Small GTPases of the Rho family modulate actin cytoskeleton remodeling to affect cell junctions [43,44]. To determine how the small GTPase Rap1 affects occludin expression, more information is required. Of note, the expression of claudin-1 was not significantly reduced by the inhibitor treatment. It may be that the effect of the inhibitor is limited or the regulation of claudin-1 is completed by the effector molecules downstream of the Rap1 signaling pathway, or there are other regulatory mechanisms for claudin-1.

In conclusion, we found that EspP2 promotes the adhesion of *G. parasuis* to host cells by up-regulating the expression of claudin-1 and occludin. Moreover, EspP2 can regulate the expression of occludin through the Rap1 signaling pathway. These findings contribute to the understanding of the pathogenic mechanism of *G. parasuis* and EspP2, and they offer improved approaches for managing *G. parasuis* infection.

## 4. Materials and Methods

### 4.1. Strains, Primers, Bacterial and Cell Culture Conditions

*G. parasuis* SH0165 was kindly supplied by Xuwang Cai from Huazhong Agricultural University and strain SC1401 was provided by the Laboratory of Research Center of Swine Disease in Sichuan Agricultural University. The culture method of each strain was as previously described [45]. Both strains were cultivated in Tryptic Soy Broth (TSB, Difco, NJ, USA) or Tryptic Soy Agar (TSA, Difco, NJ, USA) supplemented with 0.1% (*w*/*v*) nicotinamide adenine dinucleotide (NAD, Sigma-Aldrich, Rockville, MD, USA) and 5% inactivated bovine serum (Solarbio, Beijing, China) (TSB++ and TSA++, respectively). *Escherichia coli* DH5α (Biomed, Beijing, China) and BL21(DE3) (Biomed, Beijing, China) were cultured in LB broth or LB agar for protein expression. When necessary, 50 μg mL^−1^ of kanamycin or 100 μg mL^−1^ of ampicillin was added to these media. All cultures were at 37 °C with shaking at 220 rpm. Appendix A contains a list of primers used in this investigation. The new-born piglet tracheal, NPTr, was kindly supplied by Hongbo Zhou from Huazhong Agricultural University. The Laboratory of Research Center of Swine Disease at Sichuan Agricultural University preserved the porcine kidney cell line, PK15. The cells were cultured in Dulbecco’s modified Eagle medium (DMEM; Gibco, Carlsbad, CA, USA) supplemented with 10% heat-inactivated fetal bovine serum (PAN-Biotech, Aidenbach, Germany) at 37 °C in a humidified 5% CO_2_ atmosphere.

### 4.2. Construction of EspP2 Deficient Mutants

The *EspP2* deletion strain was constructed and screened as previously described [46]. Briefly, primers P1/P2 and P3/P4 were used to amplify, respectively, the 620-bp upstream and the 611-bp downstream homologous regions of *EspP2* from *G. parasuis* SC1401 genomic DNA. Next, primers P5/P6 were used to amplify a 935-bp kanamycin resistance cassette from pKD4. Subsequently, the three fragments were purified using a Qiaquick spin column kit (Qiagen, Hilden, Germany) and ligated into restriction sites *BamH I* and *Xho I* of linearized pK18mobSacB, resulting in the vector pK18-*EspP2*. A natural transformation method was used to convert pK18-*EspP2* into *G. parasuis* SC1401, as previously described [46]. After 36 h incubation, the resultant kanamycin-resistant transformants were grown to large-scale culture in TSB++ supplemented with Kan, in order to be further identified by PCR. Of note, we previously tested the natural transformation efficiency of several standard and local strains of *G. parasuis* and found that SC1401 had the highest efficiency. We therefore chose it as the wild-type strain for this study and from there constructed the *EspP2* deletion strain.

### 4.3. Expression of Recombinant EspP2

Recombinant EspP2 was expressed in *E. coli* BL21 [26]. Primers P13/P14 were used to amplify *G. parasuis EspP2* from the genomic DNA of SH0165. The PCR product was ligated to pET-32A (+) (HANBIO, Shanghai, China), resulting in the recombinant plasmid pET-*EspP2*. pET-*EspP2* was transformed into *E. coli* BL21 (DE3), protein expression was induced with 1.2 mM IPTG for 16 h at 18 °C. The induction conditions for the His-tag protein were the same as EspP2 protein. Ni-NTA His-Bind Resin (Bio-Rad, Boulder, CO, USA) was used to purify EspP2, and 2 L of PBS was used to dialyze EspP2 for 2 days at 4 °C. The purified protein was then subjected to SDS-PAGE electrophoresis and WB.

### 4.4. Adherence Assay

Adherence assays were performed as previously described with modifications for MOI [47]. The wild-type SC1401 and the *EspP2* deletion strain, Δ*EspP2*::Kan, were grown to logarithmic phase and then collected by centrifugation at 5000 rpm for 2 min. Cells were washed three times with PBS and then suspended in DMEM with 5% fetal bovine serum. Aliquoting 1 × 10^6^ PK15 cells into 6-well plate wells, bacteria were added at a multiplicity of infection (MOI) of 10 and 100. Plates were incubated at 37 °C in 5% CO_2_ for 2 h to allow for bacterial adhesion. The cells were washed 5 times with PBS to remove nonspecifically attached bacteria, then incubated in 100 μL PBS containing 0.25% trypsin for 5 min at 37 °C. Then 0.9 mL of cold TSB++ was added to each well and cells were collected by trituration. The cell suspension was 10-fold serially diluted; 100 μL from each dilution was plated onto TSA++ plates then incubated for 24 h at 37 °C. To determine the colony-forming units (CFU), the number of colonies on each plate was counted.

### 4.5. Quantitative Real-Time PCR

Refer to the previous procedure for qRT-PCR [48]. RNA from PK15 cells incubated with EspP2 was isolated using a UNIQ-10 Column Total RNA Purification Kit (Sangon, China). Two-step RT-PCR was performed using PrimeScriptTM RT Reagent Kit with gDNA Eraser (Takara, Japan). Utilizing SYBR Premix EX TaqTM II (Tli RNaseH Plus; Takara, Japan), transcripts were subjected to qRT-PCR analysis. Gene expression was quantified using the 2[-ΔΔC(T)] method, and results are presented relative to expression of *β-actin*. The Lightcycler96 (Roche, Basel, Switzerland) system was used for the qPCR. For each sample, there were three biological replicates and the qRT-PCR assay was repeated three times.

### 4.6. Western Blotting

The Western blotting assay was performed as described previously to detect the claudin-1 and occludin expression levels [49]. PK15 cells were plated into 6-well plates at a density of a 1 × 10^6^ cells/well and stimulated with EspP2 (50 μg mL^−1^) for different lengths of time. Cells were scraped off at the conclusion of each time point and placed in Eppendorf tubes. Cells were pelleted and then suspended in 100 μL of cold cell lysis solution with PMSF and incubated on ice for 30 min. Centrifugation at 12,000× *g* min^−1^ for 10 min was followed by the collection of supernatant and the addition of 25 μL of 5 × Loading Buffer. Samples were incubated for 20 min at 37 °C. Aliquots corresponding to 50 µg of each sample were subjected to 12.5% SDS-PAGE; proteins were then electroblotted onto PVDF membrane. After blocking the membrane with 5% skim milk in TBST for 2 h at room temperature (RT), the membrane was treated with rabbit anti-claudin1 mAb (1:1000), rabbit anti-occludin mAb (1:1000), or rabbit anti-ATPase Na+/K+ beta2 (1:100,000) overnight at 4 °C. After 5 TBST washes, the membrane was incubated with HRP-conjugated goat anti-rabbit IgG (1:5000) at RT for 1 h. Following incubation, TBST was used to wash the membrane 5 times. Enhanced chemiluminescence reagents (ECL; Bio-Rad, USA) were used to visualize proteins.

### 4.7. Indirect Immunofluorescence

Immunofluorescence staining of claudin-1 and occludin was used to further examine the impact of EspP2 on the subcellular distribution of these two proteins [50]. PK15 cells were grown to about 90% confluence on coverslips in a 6-well plate, 50 μg mL^−1^ of purified EspP2 was aliquoted to each well, and cells were incubated at 37 °C for 24 and 48 h. After that, the cells were washed 3 times with PBS, fixed with 4% paraformaldehyde for 15 min, and then washed once more with PBS. Cells were then blocked in PBS/BSA (1%) for 1 h at 37 °C. Cells were incubated with primary antibodies against occludin (1:1000; Abcam, MA, USA) and claudin-1 (1:1000; Abcam, MA, USA) at 4 °C overnight. Subsequently, the samples were treated for 1 h at 37 °C in the dark with fluorescein isothiocyanate (FITC) goat anti-mouse IgG (Proteintech, Beijing, China). DAPI (Beyotime, Shanghai, China) was used to label the nuclei. Results were documented using a Nikon Eclipse fluorescence microscopy (ci Series, DS-U3). Immunofluorescence analysis was performed using Case Viewer software.

### 4.8. Giemsa Staining

In addition to the manual counting of bacteria, Giemsa staining (Giemsa stain, Solarbio, Beijing, China) was also used to determine the number of bacterial adhesions. Fresh working stain was prepared by adding one-part 10× storage solution to nine-parts phosphate buffer and mixing thoroughly. Cells were air dried and then fixed with methanol for 1–3 min. Methanol was removed, enough Giemsa solution was used to cover the cells, and incubation was 15–30 min at room temperature. After a mild washing, the cells were dried in preparation for microscopic inspection [51,52].

### 4.9. Transcriptome Sequencing and Data Analysis

PK15 cells were stimulated with EspP2 protein at a final concentration of 50 μg mL^−1^ for 12 and 36 h at 37 °C. Following the manufacturer’s instructions, TRIzol reagent (Invitrogen, CA, USA) was used to extract total RNA from each sample. Novogene Bioinformatics Technology Co., Ltd. (Beijing, China) handled the RNA quantification, library preparation, clustering, and sequencing. Differential expression analysis was performed using the DEGSeq R package (version 1.20.0) [53]. The false discovery rate was managed by adjusting the *p*-values by the application of the Benjamini and Hochberg technique. Genes that showed a significant difference in expression were those with an adjusted *p*-value < 0.05. To correct for the bias associated with gene length, the GOseq R package performed a Gene Ontology (GO) enrichment analysis of differentially expressed genes [54]. GO terms were deemed significantly enriched if their corrected *p*-value was less than 0.05. Utilizing the KOBAS software (Center for Bioinformatics, Peking University and Institute of Computing Technology, Chinese Academy of Sciences), we assessed the statistical enrichment of genes that were differentially expressed in KEGG (Kyoto Encyclopedia of Genes and Genomes) pathways [55].

### 4.10. Statistical Analysis

Statistical analyses were performed using the GraphPad Prism version 6.0. Statistical significance was evaluated using Student’s *t*-test, one-way ANOVA, or two-way ANOVA. Significant differences between groups are indicated by * *p* < 0.05, ** *p* < 0.01, *** *p* < 0.001, and **** *p* < 0.0001. Error bars in all figures represent the standard deviations of three independent experiments.

## Figures and Tables

**Figure 1 ijms-25-04570-f001:**
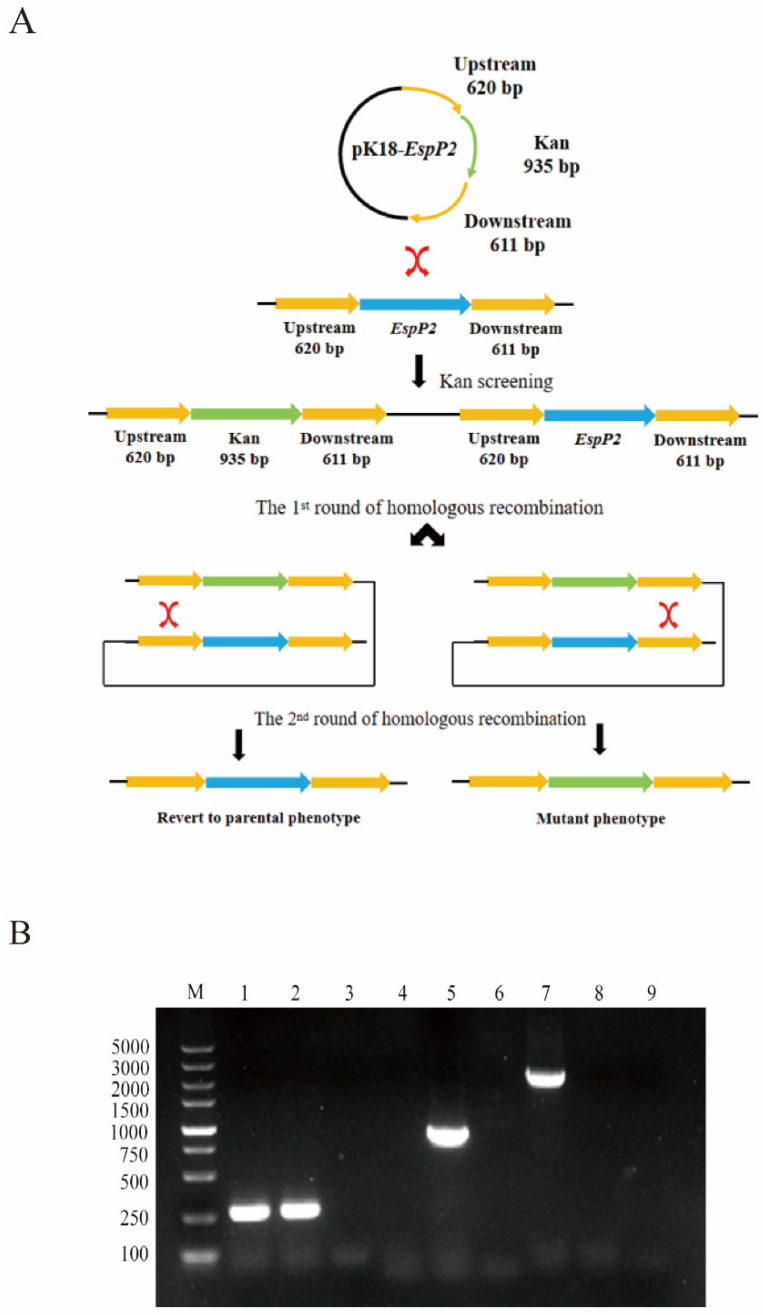
Construction and identification of the Δ*EspP2 G. parasuis*. (**A**) Schematic of the two rounds of homologous recombination used to construct Δ*EspP2 G. parasuis*. First round: homologous recombination occurs in *G. parasuis* SC1401 transformed with pK18-*EspP2*. The recipient bacterium now has both the target and resistance genes. Second round: Kan is used to replace the *EspP2* gene. (**B**) PCR identification of successful construction of Δ*EspP2*::Kan. Lane 1–3: primers P1 and P2 were utilized to amplify the species-specific marker fragment [25] of *G. parasuis* (SC1401, Δ*EspP2*::Kan, negative control). Lane 4–6: using primers P3 and P4, the kanamycin resistance cassette was amplified (SC1401, Δ*EspP2*::Kan, negative control). Lane 7–9: primers P5 and P6 were utilized to amplify the partial sequence of *EspP2* of *G. parasuis* SC1401 (SC1401, Δ*EspP2*::Kan, negative control).

**Figure 2 ijms-25-04570-f002:**
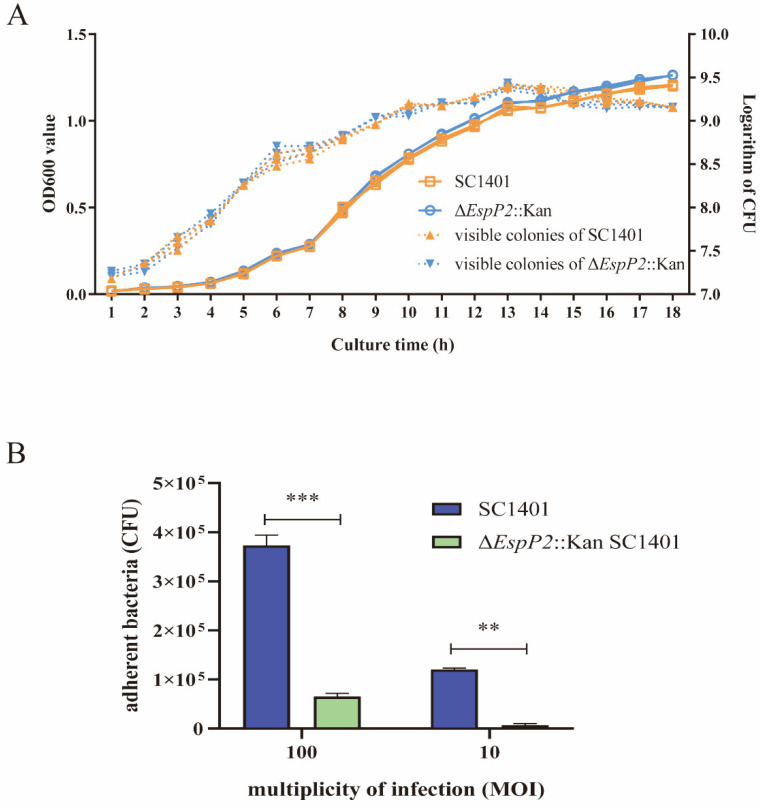
Deletion of *EspP2* decreases *G. parasuis* adherence to PK15 cells. (**A**) Growth curves of Δ*EspP2*::Kan and wild-type *G. parasuis*. OD600 and colony formation on TSA++ agar were measured over an 18 h period to track the growth of the bacteria. Three technical replicates were used to collect the data. Standard deviations are indicated by error bars. (**B**) Wild type *G. parasuis* and Δ*EspP2*::Kan adhesion to PK15 cells. The standard deviation of three separate experiments is shown by error bars. Significant differences between groups are indicated by ** *p* < 0.01 and *** *p* < 0.001.

**Figure 3 ijms-25-04570-f003:**
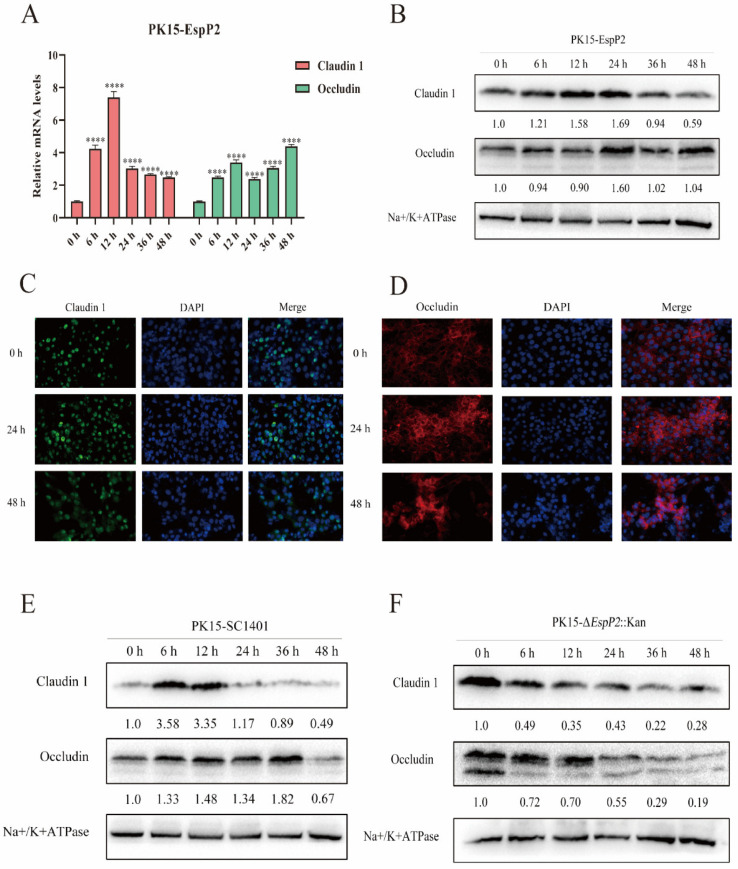
EspP2 enhances claudin-1 and occludin expression. (**A**) RT-PCR results of *claudin-1* and *occludin* levels in PK15 cells. Significant differences between groups are indicated by **** *p* < 0.0001. (**B**) Representative Western blot of claudin-1 and occludin in EspP2-treated PK15 cells. (**C**,**D**) Indirect immunofluorescence of PK15 cells treated with EspP2 showing claudin-1 (green), occludin (red), and nuclei (blue) are stained with DAPI. PK15 cells infected with (**E**) wild-type SC1401 and (**F**) Δ*EspP2*::Kan at a MOI 10; cells were collected at 0, 6, 12, 24, 36, and 48 hpi, whole-cell extracts were prepared, and levels of occludin and claudin-1 were detected using Western blot.

**Figure 4 ijms-25-04570-f004:**
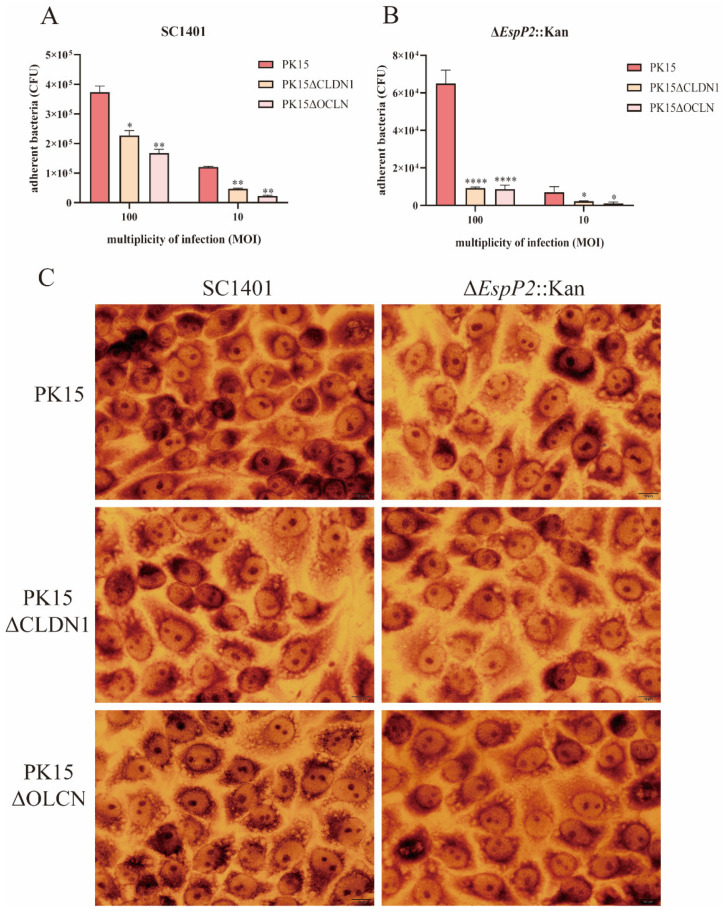
EspP2 regulates adhesion by affecting claudin1 and occludin. (**A**) Adherence of wild-type SC1401 to PK15, claudin-1 knockout, and occludin knockdown cells. The standard errors from three separate experiments, each with three copies of each sample, are represented by error bars. Significant differences between groups are indicated by * *p* < 0.05, ** *p* < 0.01. (**B**) Adherence of Δ*EspP2*::Kan to PK15, claudin-1 knockout, and occludin knockdown cells. Significant differences between groups are indicated by * *p* < 0.05, **** *p* < 0.0001. (**C**) Bacterial adhesion as detected by Giemsa stain. After incubating the cells for two hours with SC1401 and Δ*EspP2*::Kan, the cells were washed and stained with Giemsa.

**Figure 5 ijms-25-04570-f005:**
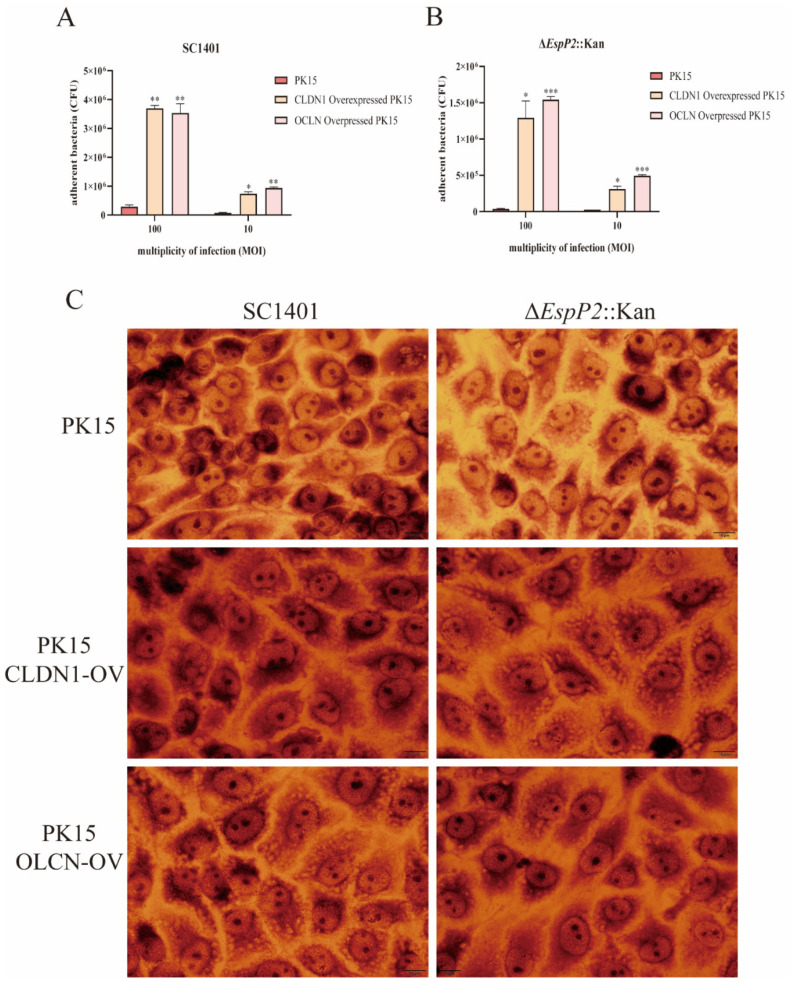
*G. parasuis* adhesion is increased in claudin-1 and occludin overexpressing cells. (**A**) Adherence of wild-type SC1401 to PK15, claudin-1 and occludin overexpressing cells lines. The standard errors from three separate experiments, each with three copies of each sample, are represented by error bars. Significant differences between groups are indicated by * *p* < 0.05 and ** *p* < 0.01. (**B**) Adherence of Δ*EspP2*::Kan to PK15, claudin-1, and occludin overexpressing cells lines. Significant differences between groups are indicated by * *p* < 0.05 and *** *p* < 0.001. (**C**) Bacterial adhesion to three cell lines as detected by Giemsa stain. After incubating the cells for two hours with SC1401 and Δ*EspP2*::Kan, the cells were washed and stained with Giemsa.

**Figure 6 ijms-25-04570-f006:**
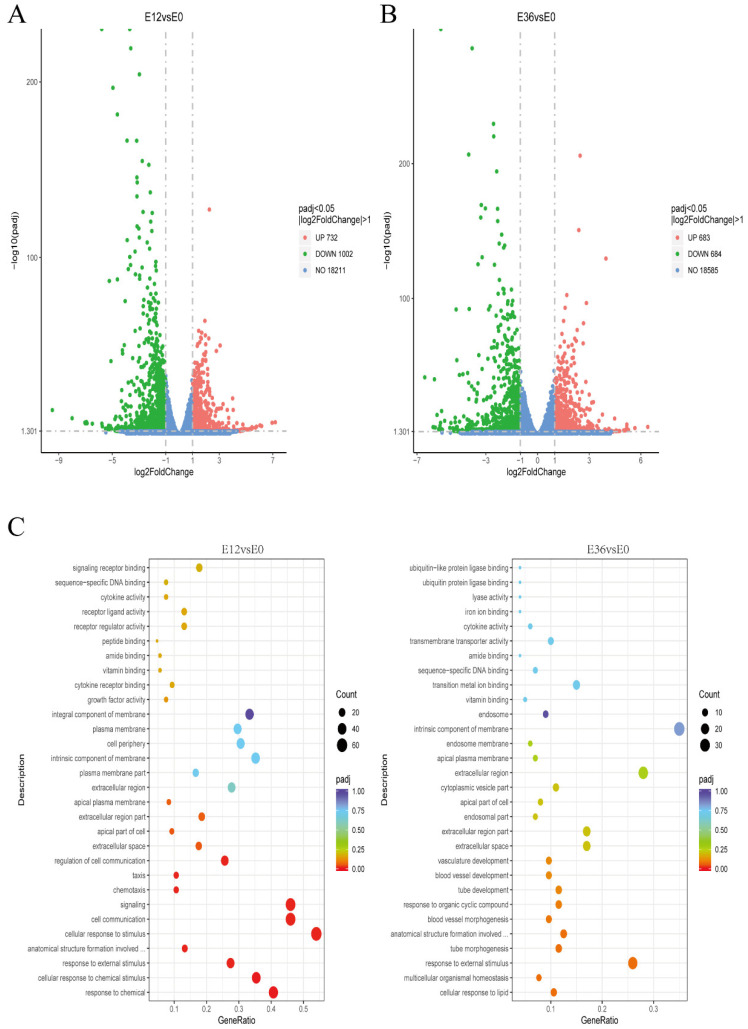
Volcano map and GO enrichment analysis. Volcano plot of differentially expressed genes between 0 and (**A**) 12 h, and (**B**) 36 h. Red dots indicate genes that are significantly up-regulated, green dots indicate genes that are significantly down-regulated, and blue dots indicate genes that are not differently expressed. (**C**) The abscissa is the ratio of the number of differentially expressed genes annotated to the GO term to the total number of differentially expressed genes; the ordinate is the enriched GO term. The degree of enrichment is indicated by the color depth.

**Figure 7 ijms-25-04570-f007:**
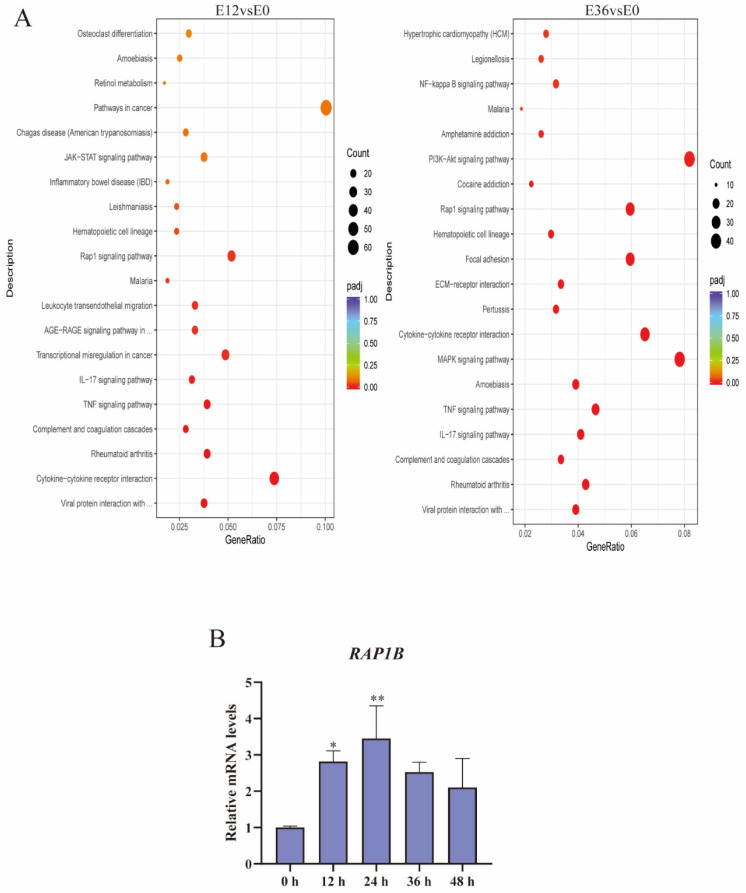
Analysis of KEGG pathway enrichment. (**A**) The abscissa is the ratio of the number of differentially expressed genes annotated to the KEGG pathway to the total number of differentially expressed genes. Pathway names are on the ordinate. The number of genes identified in the KEGG pathway is represented by the size of the point. The significance of enrichment is represented by the color spectrum from red to purple. (**B**) The transcription level of *RAP1B*. Significant differences between groups are indicated by * *p* < 0.05, ** *p* < 0.01.

**Figure 8 ijms-25-04570-f008:**
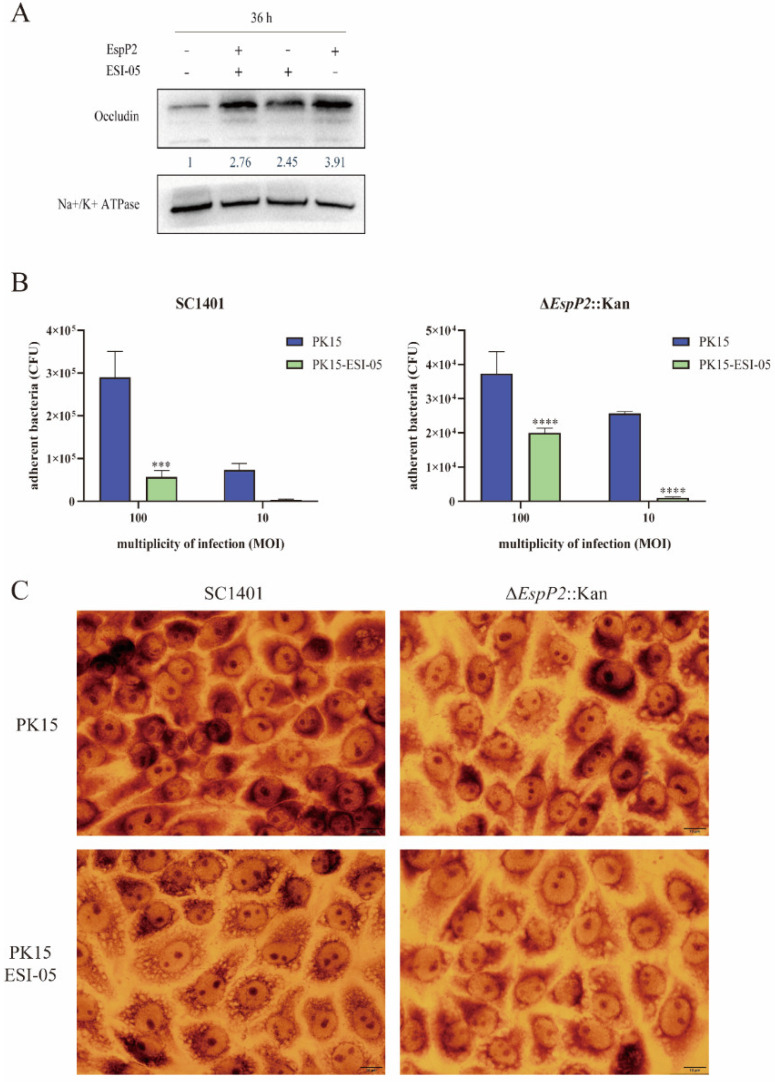
Inhibition of Rap1 signaling reduces occludin expression and inhibits *G. parasuis* adhesion. (**A**) Representative Western blot of occludin in PK15 cells treated with inhibitor ESI-05 for 36 h. (**B**) Adhesion of wild-type and Δ*EspP2*::Kan to PK15 cells treated with inhibitor. The standard deviation of three separate experiments is shown by error bars. Significant differences between groups are indicated by *** *p* < 0.001 and **** *p* < 0.0001. (**C**) Giemsa stain of PK15 cells treated with inhibitor and incubated with wild-type and Δ*EspP2*::Kan for 2 h.

## Data Availability

The data presented in this study are available upon request from the corresponding author.

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
