# Peer review of "EspP2 Regulates the Adhesion of Glaesserella parasuis via Rap1 Signaling Pathway"

_ijms, 2024, doi:10.3390/ijms25084570_

Round 1

Reviewer 1 Report

Comments and Suggestions for Authors

Dear Authors,

Please see my suggestions below:

Please edit E. coli  with a blank between genus and species name and italicized in lines 52 and 54

Please support the study methodology with appropriate references in the method section. The authors only give reference in 4.2, 4.4, and 4.9, but other parts are left unsupported

I think you should write more original sentences because the manuscript has a high similarity rate. The study has valuable content and results showed more clearly. Thus using a more original language will keep you from plagiarism. 

Reviewer 2 Report

Comments and Suggestions for Authors

Tang et al. demonstrated the effect of EspP2 on virulence by using  G. parasuis SC1401 in its wild-type form and SC1401 that was made EspP2-deficient. They showed that EspP2-deficiency resulted in significantly reduced adhesion of G. parasuis to PK15 cells, and also that EspP2 causes up-regulation of claudin-1 and occludin expression. The topic is interesting and important. This study contributes to the understanding of the mechanisms of adhesion by G. parasuis, and provides new insights for the development of new drugs. Please see the suggestions below.

  1. The loss of EspP2 results in reduced bacterial adhesion PK-15 epithelial cells. Did the authors tried to test the adhesion using other cells?
  2. Rap 1 signaling pathway was analyzed. The authors need to clarify what is the rational to select this pathway. The author used ESI-05 to inhibit the Rap1 pathway, will this inhibitor also affect other pathways? It is still not clear how this pathway affects occluding expression.
  3. The discussion is not well written as this part fails to elucidate the mechanism behind the findings but focus on the results. Line 281- line 295 still focus on the findings of the study.
  4. more details are needed in the M&M section. for example, " appropriate primary (1:100) and the secondary antibodies", what is the appropriate antibody?
  5. English editing is needed. 
Comments on the Quality of English Language
  1. English editing is needed. 

Round 2

Reviewer 2 Report

Comments and Suggestions for Authors

this is an updated version; the authors have modified the manuscript according to the suggestions. the manuscript is improved I did not detect concerns.